# Simulation Study on Weld Formation in Full Penetration Laser + MIG Hybrid Welding of Copper Alloy

**DOI:** 10.3390/ma13235307

**Published:** 2020-11-24

**Authors:** Feipeng An, Qilong Gong, Guoxiang Xu, Tan Zhang, Qingxian Hu, Jie Zhu

**Affiliations:** 1Luoyang Ship Material Research Institute, Luoyang 471023, China; anfeipeng@163.com; 2School of Materials Science and Engineering, Jiangsu University of Science and Technology, Zhenjiang 212003, China; 13773726659@163.com (Q.G.); z476032818@163.com (T.Z.); huqingxian@126.com (Q.H.); zhujie_5858@163.com (J.Z.)

**Keywords:** full penetration hybrid welding, copper alloy, weld pool behavior, fluid flow, numerical simulation

## Abstract

Considering the coupling of a droplet, keyhole, and molten pool, a three-dimensional transient model for the full penetration laser + metal inert gas (MIG) hybrid welding of thin copper alloy plate was established, which is able to simulate the temperature and velocity fields, keyhole behavior, and generation of the welding defect. Based on the experimental and simulation results, the weld formation mechanism for the hybrid butt welding of a 2 mm-thick copper alloy plate was comparatively studied in terms of the fluid dynamic feature of the melt pool. For single laser welding, the dynamic behavior of liquid metal near the rear keyhole wall is complex, and the keyhole has a relatively drastic fluctuation. An obvious spattering phenomenon occurs at the workpiece backside. Meanwhile, the underfill (or undercut) defect is formed at both the top and bottom surfaces of the final weld bead, and the recoil pressure is identified as the main factor. In hybrid welding, a downward fluid flow is strengthened on the rear keyhole wall, and the stability of the keyhole root is enhanced greatly. There are large and small clockwise vortexes emerging in the upper and lower parts of the molten pool, respectively. A relatively stable metal bulge can be produced at the weld pool backside. The formation defects are suppressed effectively, increasing the reliability of full penetration butt welding of the thin copper alloy plate.

## 1. Introduction

Copper and its alloys have been widely used in power, chemical, shipbuilding, and heat exchange industries due to the excellent properties, which concern various kinds of thin wall welded structures. However, some of their properties also make copper and its alloys difficult to weld [1,2,3,4,5]. The major issues in conventional arc welding include the difficulty of the melting material, large residual stress and distortion, wider heat-affected zone (HAZ), low welding efficiency, and so on [2,4]. The laser welding of copper and its alloy is also a challenging task owing to their high reflectivity to light, for which a strict assembly requirement of specimen is needed [5,6,7], and the surface underfill at the weld top side and root is difficult to control during the autogenous full penetration welding process [8,9]. As one of the most promising technologies, laser+gas metal arc welding (GMAW) hybrid welding combines the advantages of each component and overcomes their limitations, which has great potential to realize the high quality and high efficiency welding of a thin plate, thus having received more and more attention in the joining of copper [1,2]. However, hybrid welding technology involves a large number of welding parameters. Only when the matching of all these parameters meets a certain demand can an acceptable weld bead be obtained [1,4,5], leading to relatively large difficulty in process optimization. Thus, a deep understanding of the physical mechanisms is essential to improve the stability and reliability of full penetration hybrid butt welding quality.

Currently, the studies on the hybrid welding of copper and its alloy depend on the experimental observations heavily, a few of which involve the internal physical mechanism for full penetration hybrid welding. Through the experimental method, Zhang et al. [2,4] optimized the welding process and analyzed the effects of different parameters on the weld geometry and microstructures for single-pass laser + MIG hybrid welding of a pure copper plate without preheating. Gong et al. [10] also experimentally investigated the microstructure and mechanical properties of a laser-arc hybrid butt-welded joint for 2 mm thick pure copper. They found that the heat input as well as process parameters had no obvious effect on the ultimate tensile strength of the joint but could affect its elongation largely. With the aid of a high-speed imaging system, Yang et al. [11] analyzed the arc and weld pool behaviors in full penetration hybrid butt welding of 8 mm-thick pure copper and studied the influence of process parameters on the joint geometry and integrity. However, their work did not concern the effect of keyhole and fluid flow within the melt pool owing to the limitation of the detection method. Wang et al. [1] also adopted a high-speed camera to observe the dynamic behaviors of the molten pool, keyhole, and droplet transfer in laser-arc hybrid welding of pure copper. They concluded that the process stability of hybrid welding was attributed to the two key issues, i.e., keyhole stability and the avoidance of arc wandering. Although these experimental results can improve the qualitative understanding of the hybrid welding process to some extent, they cannot reveal the internal physical mechanism of hybrid welding for copper and its alloy completely.

To make up for the deficiency in the experimental methods, researchers have also conducted lots of simulation studies on the mechanism of laser + GMAW hybrid welding [12,13,14,15,16,17,18,19,20], thus leading to significant advances in the fundamental understanding of the hybrid welding process. However, previous efforts are mainly focused on the partial penetration welding of steel or aluminum alloy, and there are very limited studies in the literature involving the analysis of full-penetration hybrid welding [21,22], especially for copper alloy. With advancements in the commercial software, more accurate models for hybrid welding have been reported. Cho et al. [12] proposed a three-dimensional model based on a FLOW-3D commercial code, which considered the coupling of a droplet, keyhole, and weld pool and the real-time multi-reflections of a laser beam. Using this model, they studied the fluid flow feature in bead-on-plate hybrid welding of mild steel. However, this model is time consuming. Wu et al. [13] established a model through FLUENT software and quantitatively analyzed the transient evolution of fluid flow and temperature field in a hybrid weld pool with a dynamic keyhole. However, their studies were also limited to the partial penetration bead-on-plate welding of stainless steel. In order to enhance the calculation efficiency, Xu et al. [14] established the adaptive volumetric heat source model to describe the laser energy. They numerically studied the fluid flow phenomenon and the formation process of keyhole-induced pore in partial penetration hybrid welding of aluminum alloy with a model considering the coupling of gas, liquid, and solid phases. Recently, Chen et al. [20] proposed a unified model for laser-arc hybrid welding, which incorporated the dynamic coupling of a keyhole, metal vapor, arc plasma, and welding. However, their work still only concerned the bead-on-plate welding process of stainless steel. Meanwhile, due to complexity, the model is more suitable for investigating the interaction of arc, keyhole, and melt pool, not the weld formation. Farrokhi et al. [21] built a finite element model for full penetration and partial penetration hybrid laser welding of thick-section steel and proposed a new double-conical volumetric heat source. However, their study was only focused on the temperature field, and no fluid flow was concerned. A similar work was done for full penetration laser welding by Kazemi et al. [23]. Chen et al. [24] numerically investigated the influence of magnetic field orientation on the basic velocity distribution in full penetration laser butt welding of aluminum alloy. Nevertheless, for their model, the keyhole geometry and size was set before simulation and also did not allow for the free surface distortion of molten pool. By means of the numerical model of Cho et al. [12], Zhang et al. [25] also analyzed the effect of a joint gap on weld bead formation in laser butt welding of stainless steel in terms of the dynamics of keyhole, molten pool, and laser-induced plume. A similar model was utilized by Wu et al. [26], who simulated the spatter formation in the laser welding of aluminum alloy at the full penetration position.

A general review of previous research studies demonstrates that although lots of related simulation efforts have been made, the research results are not applicable to full penetration hybrid welding of the copper alloy due to large differences in the material properties, joint geometry, or welding technology. Up to now, there is still a lack of useful information on the internal mechanism for the hybrid butt welding of copper alloy in the full penetration mode. Therefore, the purpose of this study is to establish a unified three-dimensional transient numerical model to study the weld pool dynamic behavior in full-penetration laser + MIG hybrid butt welding of copper alloy. The fluid flow phenomenon within the molten pool is numerically investigated, and the influence of arc power on weld formation is also analyzed, which contributes to an improved understanding of the physical and formation mechanism in full penetration hybrid butt welding of copper alloy and the selection of process parameters.

## 2. Experimental

Fiber laser + MIG hybrid butt welding experiments were performed on an H62 copper alloy plate with a thickness of 2 mm under different welding conditions. The filler material is S221 copper alloy wire with a diameter of 1.2 mm. The chemical compositions of the welding wire are listed in Table 1. As shown in Figure 1, during the hybrid welding process, a fiber laser heat source with a 6 kW maximum power is employed, which is in front of the arc. To avoid the possible damage of optical components by reflected light, the laser beam is inclined 5° forwards. It has a 1.07 μm wavelength, 0.3 mm focal spot diameter, and 300 mm focal length. The focal position in welding is set at 0 mm. The axis of the MIG torch is tilted 20° relative to the centerline of the laser beam. The distance between the laser and the arc on the surface of the workpiece is 1 mm. The shielding gas is 100% Ar, and the flow rate is 20 L/min. The joint gap takes 0 mm. Other parameters are listed in Table 2.

## 3. Modeling

In this section, the mathematical models involving the heat input, force source, and boundary condition are introduced. The related solution method is also given.

### 3.1. Governing Equations

Similar to the work by Cho et al., both gas and metallic liquids are assumed as laminar, incompressible, and Newtonian fluids in this study. Vapor plume and shielding gas in the keyhole are also neglected. The pressure in the keyhole is assumed to be atmospheric pressure. Thus, the governing equations of energy, momentum, and mass can be expressed as follows.

Energy conversation equation:(1)$$\frac{\partial (\rho H)}{\partial t}+V\cdot \nabla (\rho H)=\nabla \cdot (k\nabla T)+S_{V}$$

Momentum conversation equation:(2)$$\rho \left(\frac{\partial V}{\partial t}+V\cdot \nabla V\right)=-\nabla p+\mu \nabla^{2}V-\frac{\mu V}{K}+F_{b}+F_{em}$$

Mass conversation equation:(3)$$\frac{\partial \rho }{\partial t}+\nabla \cdot (\rho V)=S_{m}$$
where *ρ* is the density; *t* is the time; *H* is the enthalpy; *k* is the thermal conductivity; *V* is the velocity vector; *p* is the pressure; *g* is the gravitational vector; *T_ref_* is reference temperature; *K* is the drag coefficient in the porous media; *β* is the thermal expansion coefficient; *F_em_* is the electro-magnetic force vector; *μ* is the viscosity; *S_m_* represents the mass source term. *S_v_* denotes the heat source term, which involves both arc and laser heat inputs. The third term at the right side of Equation (2) is the source term due to the frictional dissipation in the mushy zone; *F_b_* is the buoyancy force, which is calculated as
(4)$$F_{b}=\rho g\beta \left(T-T_{ref}\right)$$
where *g* is the gravitational vector; *T_ref_* is the reference temperature;

In calculation, the enthalpy due to the variation between the solid and liquid phases should be taken into account, which is dealt with using the enthalpy-porosity method in this study. Thereby, the total enthalpy can be determined by the following equation [27]
(5)$$H=\int c_{p}dT+f_{l}L_{m}$$
where *c_p_* is the specific heat; *L_m_* is the latent heat of fusion; *f_l_* is the liquid fraction, which changes linearly with temperature and is defined in [26]; the first and second terms at the right side represent the sensible heat and latent heat content, respectively.

To simplify the model, the Carman Kozeny equation is used to estimate the drag coefficient, which is given by [28]
(6)$$K=\frac{f_{l}^{3}d^{2}}{180{\left(1-f_{l}\right)}^{2}}$$
where *d* is a constant in the order of 10^−4^ m.

In hybrid welding, the mass exchange of liquid and gas phases occurs at the gas–liquid interface due to laser-induced evaporation, and a simple mass source is considered in the mass continuity equation [29,30].
(7)$$S_{m}=\{\begin{array}{c} m_{er}\\ -m_{er} \end{array}$$
where *m_er_* is the evaporation rate.

### 3.2. Heat Source Model

In full penetration hybrid welding, the weld pool free surface has a relatively large depression. Thus, it is reasonable to model the arc heat input as a double elliptical heat source [31], which is given by
(8)$$q_{a}=\frac{\eta_{A}IU}{\pi \left(a_{f}+a_{r}\right)b_{h}c_{h}}exp\left(-\frac{3{\left(x-v_{0}t\right)}^{2}}{a_{f}^{2}}-\frac{3y^{2}}{b_{h}^{2}}-\frac{3z^{2}}{c_{h}^{2}}\right)\;x⩾0\;x\geq 0$$
(9)$$q_{a}=\frac{\eta_{A}IU}{\pi \left(a_{f}+a_{r}\right)b_{h}c_{h}}exp\left(-\frac{3{\left(x-v_{0}t\right)}^{2}}{a_{r}^{2}}-\frac{3y^{2}}{b_{h}^{2}}-\frac{3z^{2}}{c_{h}^{2}}\right)\;x<0\;x\geq 0$$
where *v*_0_ is the welding speed; *I* is the welding current; *U* is the arc voltage; $\eta_{A}$ is the arc efficient; $a_{r}$, $a_{f}$, $b_{h}$ and $c_{h}$ are the distribution parameters.

In laser welding, due to the high density of the focused beam, a keyhole is generated, so that more laser heat is able to act inside the workpiece directly. For the fiber laser beam, the influence of the IB absorption is small because of the short wavelength, and the Fresnel absorption is mainly due to multiple reflections of the laser beam in the keyhole. Some researchers have proposed several consistent keyhole models [28,32,33], which allow for the multiple Fresnel absorption of laser energy using ray tracing technology. However, these models are very time consuming and involve more unknown material properties. In this study, a curve-rotated volumetric heat source model based on the small hole size and adjustable heat flow peak is used to describe the laser heat input, which is expressed as follows [14].
(10)$$q_{L}=\frac{3\eta_{L}Q_{L}}{\pi \left(1-e^{-3}\right)(a+b)}\left[\frac{(1-\chi )z_{i}z_{e}}{z_{i}-z_{e}}\frac{1}{z}+\frac{\chi z_{i}-z_{e}}{z_{i}-z_{e}}\right]exp\left\{-\frac{3\left(x^{2}+y^{2}\right)}{{\left[r_{0}\left(z\right)\right]}^{2}}\right\}$$
(11)$$r_{0}\left(z\right)=\frac{r_{e}-r_{i}}{z_{e}^{2}-z_{i}^{2}}z^{2}+\frac{r_{1}z_{e}^{2}-r_{e}z_{i}^{2}}{z_{e}^{2}-z_{i}^{2}}$$
where *η_l_* is the laser power efficient; *Q_L_* is the laser power; *r_e_* and *r_i_* are the radii of the heat source top and bottom surfaces, respectively; *χ* represents the proportion coefficient between peak power densities at the top and bottom surfaces of the heat source, which is set at 1.5; *z_e_* and *z_i_* are z-coordinates of the heat source top and bottom surfaces, respectively. In calculation, the geometric parameters (i.e., *r_e_*, *r_i_*, *z_e_*, and *z_i_*) are determined according to the size of the keyhole, which changes with time *t*. Based on the work of Zhao et al. [29], the power efficiency of the laser is estimated by a simplified equation. When the keyhole appears, the efficiency is arbitrarily enhanced by about 8% in order to compensate for the laser energy distributed outside the melt pool. Meanwhile, different from that in partial penetration welding, in full penetration hybrid welding, part of the laser beam leaves the keyhole from the keyhole exit at the workpiece bottom, leading to the decrease of laser thermal efficiency to some extent. In the case of a penetrated keyhole, the efficiency is again according to the experimental results.

### 3.3. Droplet Model

In MIG, the tip of the wire is heated by arc welding to form a high-temperature droplet, which grows continuously and eventually impacts the free surface of the molten pool at a certain speed, affecting the heat flux distribution and fluid flow in the molten pool. To decrease the time cost of calculation, this phenomenon is depicted with a simplified model. In calculation, the droplet transfer is regarded as the process in which the liquid metal flows into the molten pool from a certain area above the molten pool at a certain speed and period, as illustrated in Figure 2. The initial velocity of the droplet is estimated by [34].
(12)$$v_{d}=\sqrt{0.33692+0.00854\left(I/2r_{d}\right)}$$
where *r_d_* is the droplet radius.

The droplet is assumed to be spherical and has the same diameter as the wire diameter, with an initial temperature of 2200 K. The droplet transfer frequency is determined by the wire feed speed and the droplet diameter. Except for gravity, metal droplets are also subjected to drag force by arc plasma, and it is an estimated value by [35]
(13)$$F_{d}=c_{d}\pi r_{d}^{2}\left[\frac{\rho_{0}{\left(k_{c}I\right)}^{2}}{2}\right]$$
where *c_d_* is the drag coefficient; *ρ*_0_ is the plasma density, which is taken as the density of Argon shielding gas; and *k_c_* is the calculation coefficient, which is set at 0.5.

### 3.4. VOF Method

In this paper, the volume of fluid (VOF) method is adopted to track the gas–liquid interface. The liquid surface configuration is defined by the volume fraction function *F*, which is determined by the following equation
(14)$$\frac{\partial F}{\partial t}+u\frac{\partial F}{\partial x}+v\frac{\partial F}{\partial y}+w\frac{\partial F}{\partial z}=0$$
where *u*, *v*, and *w* are the fluid velocities in the *x*, *y*, and *z* directions, respectively.

### 3.5. Boundary Conditions

As seen in Figure 2, plane AB is set as the velocity inlet, where the liquid metal enters into the weld pool to simulate the droplet transfer. The planes AE, BF, HD, and GC are the pressure outlet, and other boundaries are set as the wall.

Due to that both the laser and arc work on the weld pool top surface, the energy boundary condition [14] at this location are given as
(15)$$-k\partial T/\partial n=q_{arc}+q_{laser}-q_{c}-q_{r}-q_{e}$$
where *q_arc_* and *q_laser_* are the arc and laser heat inputs, respectively. *n* is a vector perpendicular to the local surface, and its direction is toward the inside of the molten pool; *q_c_*, *q_r_*, and *q_e_* represent the heat losses due to convection, radiation, and evaporation, respectively; they are expressed as follows.
(16)$$q_{c}=\alpha \left(T-T_{0}\right)$$
(17)$$q_{r}=\varepsilon \sigma \left(T^{4}-T_{0}^{4}\right)$$
(18)$$q_{e}{=m}_{e}L_{b}$$
where *α* is the convective heat transfer coefficient; *ε* is the surface radiation emissivity; *σ* is the Stefan–Boltzmann constant; *m_e_* is the evaporation rate; *L_b_* is the latent heat of evaporation; and *T*_0_ is the ambient temperature.

For the weld pool bottom surface, there is no electric arc working, and the boundary condition is given by
(19)$$-k\partial T/\partial n=q_{laser}-\alpha \left(T-T_{0}\right)-\varepsilon \sigma \left(T^{4}-T_{0}^{4}\right)-m_{e}L_{b}.$$

For the other boundaries, the energy boundary condition is expressed as follows [30]:(20)$$-k\partial T/\partial n=-\alpha \left(T-T_{0}\right)-\varepsilon \sigma \left(T^{4}-T_{0}^{4}\right).$$

In hybrid welding, the keyhole has a strong stirring effect on the fluid in the weld pool, thereby affecting the dynamic behavior of the melt pool greatly. In this study, the keyhole is regarded as the surface deformation of the molten pool under the combined action of laser-induced steam reaction force, arc pressure, and surface tension. The pressure boundary condition along the normal direction of the free surface is given as [14]
(21)$$P=P_{A}+P_{R}+P_{D}-P_{S}+2\mu \frac{\partial V_{s}}{\partial s}$$
where *P_A_* is the arc pressure; *P_R_* is the recoil pressure; *P_S_* is the surface tension. *P_D_* is the droplet impingement force, which is determined by the droplet transfer model in the calculation.

The arc pressure adopts a double elliptic distribution mode, and its expression is given as follows [30]
(22)$$P_{A}(x,y)=C\frac{3\mu_{0}I^{2}}{2\pi^{2}\left(a_{i1}+a_{i2}\right)b_{i}}exp\left(-\frac{3{\left(x-v_{0}t\right)}^{2}}{a_{i1}^{2}}-\frac{3y^{2}}{b_{i}^{2}}\right)(x⩾0)$$
(23)$$P_{A}(x,y)=C\frac{3\mu_{0}I^{2}}{2\pi^{2}\left(a_{i1}+a_{i2}\right)b_{i}}exp\left(-\frac{3{\left(x-v_{0}t\right)}^{2}}{a_{i2}^{2}}-\frac{3y^{2}}{b_{i}^{2}}\right)(x<0)$$
where *μ*_0_ is the permeability; *C* is the calculation coefficient; *a_i_*_1_, *a_i_*_2_, and *b_i_* are the arc pressure distribution parameters.

The recoil pressure is the main driving force for generating the keyhole, and its calculation formula is given as follows [16]:(24)$$P_{R}=0.54P_{0}exp\left[L_{h}\left(T-T_{b}\right)/\left(RTT_{b}\right)\right]$$
where *P*_0_ is the atmospheric pressure; *L_h_* is the latent heat of evaporation; *R* is the universal gas calculation constant; *T* is the temperature of the weld pool free surface; *T_b_* is the boiling temperature of the copper alloy.

The surface tension can be expressed as [18,28]
(25)$$P_{S}=k_{c}\gamma$$
where *k_c_* is the free surface curvature; *γ* is the surface tension coefficient.

The Marangoni shear stress on the molten pool surface is tangent to the free surface, which is estimated by the following analytical solution
(26)$$\mu_{s}\frac{\partial V_{s}}{\partial s}=-\frac{\partial \gamma }{\partial T}\frac{\partial T}{\partial s}$$
where *V_s_* is the tangential component of velocity; *s* is the tangential vector of local surface.

The shear drag force caused by the arc plasma jet also has an impact on the behavior of the weld pool, which is estimated by the following analytical solution [33]
(27)$$\tau_{p}=\frac{\rho_{a}v_{a}^{2}g\left(r_{r}/H_{a}\right)}{\sqrt{R_{0}}{\left(H_{a}/D_{n}\right)}^{2}}$$
where *ρ_p_* is the density of the arc; *g_2_* is the universal function; *H_a_* is the nozzle height, which is taken as arc length; *D_n_* is the diameter of welding wire; *R*_0_ is the Reynolds number.

### 3.6. Calculation Method

The hybrid butt welding process of copper alloy was simulated by FLUENT software. The gas, liquid, and solid phases are solved in the same computational domain, and the boundary conditions of the two-phase interface are considered to be volume control equations [36]. The pressure-implicit operator split (PISO) scheme is utilized to perform the coupling of the temperature and velocity fields. A non-uniform grid system is applied. The smallest grid is 0.1 mm. At the same time, the variable time steps are used. The minimum time step is 10^−6^ s. Table 3 provides the thermo-physical and other parameters used in calculation.

## 4. Results and Discussions

Figure 3 shows the temperature and velocity fields at the longitudinal sections of the molten pool for different times at 0 A welding current (i.e., single laser welding). It can be seen that owing to the concentration of laser energy and extremely high welding velocity as well as large thermal conductivity, the weld pool has a small size. The keyhole is obviously bent backwards, leading to the laser energy mainly acting on the front wall of the keyhole. The liquid metal layer of the keyhole front wall is thicker than that of its rear wall, and the peak of the fluid flow velocity also appears in the region, which reaches 5.65 m/s. Under the action of recoil pressure, the liquid metal near the keyhole front wall always flows downwards. However, for the rear wall of the keyhole, the molten metal has more complex dynamic behavior, which will be discussed in the following section. It can be clearly observed that the keyhole depth has a quite strong fluctuation, i.e., the blind hole and through-hole modes alternate during the welding process, thereby causing the instability of the weld root and the resulting bottom formation.

In Figure 3a, it is seen that at *t* = 0.0102 s, a small molten pool is formed, in which a shallow and narrow keyhole also occurs. At this time, the liquid metal on both the front and rear walls of the keyhole flows downwards with high velocity due to the influence of recoil pressure. As the time goes on, the molten pool volume and keyhole size increase rapidly. In addition, the flow pattern of the molten metal near the rear wall keyhole has a change, which moves upwards owing to the push of fluid from the keyhole front wall and the surface tension, as indicated in Figure 3b,c. At this moment, the recoil pressure has a relatively minor effect on the keyhole rear wall because of the relatively big width of the keyhole. The laser beam cannot irradiate the rear wall directly, which can exert its influence only through its multiple reflections. In addition, similar to that at the keyhole front wall, part of the liquid metal at the keyhole bottom flows downwards driven by recoil pressure. At *t* = 0.0121 s, a fully penetrated molten pool is generated, as seen in Figure 3d. Due to the effect of gravity and recoil pressure, a liquid metal bugle happens at the workpiece bottom surface, which decreases the constraint of the liquid metal pool by the solidus phase in the thickness direction to some extent. Meanwhile, the influence of the momentum from the fluid at the keyhole front wall on the rear wall behavior is also reduced, thus leading to the liquid metal at the lower part of the keyhole rear wall flowing downward under the effect of gravity. After a very short time, the keyhole penetrates the weld pool completely, as illustrated in Figure 3e,f. Due to the high welding velocity, the keyhole lower part bends backwards severely, and the liquid metal near the keyhole exit also travels backwards at a high velocity, making the weld pool length at the workpiece bottom be close to or even larger than that at the top surface, which is consistent with the simulation work of Powell et al. [8]. Meanwhile, the spatter occurs at the bottom surface of the workpiece under the combined action of large recoil pressure and strong downward momentum of fluid, as observed in Figure 3g. Part of the liquid metal escapes from the molten pool after overcoming the surface tension, leading to a direct reduction in the liquid metal volume. This phenomenon is mainly responsible for the generation of an undercut defect on the weld root. In addition, it causes the occurrence of the surface underfilling (or depression) defect at the weld bead top side. These calculated results agree with the experimental observations of Zhang et al. [9], thus further validating the accuracy of the built model.

However, it should be noted that in the case of the fully penetrated keyhole, the distribution mode of the recoil pressure caused by evaporation has significant variation, which is reduced sharply near the keyhole exit on the molten pool bottom surface. Consequently, the keyhole tends to be closed, owing to the action of the surface tension of liquid metal, as seen in Figure 3i,j. Then, the recoil pressure distribution feature changes again. Therefore, the opening and closure of the keyhole takes place alternately at the back of the workpiece in full penetration laser welding, resulting in a severe vibration of the weld pool root. In Figure 3l,m (the latter is the zoomed-in weld pool), it is illustrated that as expected, the depression and ditch defects emerge at the bead top and bottom surfaces, respectively, at *t* = 0.085 s, which can be found in the calculated results of the weld cross-section, also indicating that the developed model can simulate the welding process reasonably.

In Figure 4, it is revealed that due to the high welding speed and large recoil pressure, the liquid metal layer near the keyhole edges is quite thin. At the middle and lower parts of the keyhole, the molten metal flows downwards with high velocity. At the region near the upper part of the keyhole, the recoil pressure has a relatively small effect owing to the relative size of the keyhole opening and the liquid metal moves upwards driven by the surface. Similar to that shown in Figure 3, a spatter phenomenon can be observed after the keyhole penetrates the molten pool fully, as seen in Figure 4e–h. In Figure 4i, it is seen that with the heat source moving forwards, the keyhole is filled by the molten metal. While the liquid metal pool is completely solidified, a depression of the weld top surface occurs, and a ditch is also generated at the weld bead back owing to the volume shrinkage of the liquid metal. Here, it should also be pointed out that during the welding process, the temperature and velocity profiles are not symmetrical strictly due to its complex dynamic behavior. Meanwhile, the solidification of the weld pool at the workpiece surface is later than that at the workpiece top surface, as shown in Figure 4j. This is because the melt pool at the workpiece bottom is behind that at the workpiece top surface.

Figure 5 gives the evolution of temperature and velocity fields at the longitudinal section in full penetration hybrid welding. As stated above, due to the large fluidity of molten copper alloy, it is difficult to obtain the satisfied weld reinforcement during MIG in the case of the extremely high welding speed. Thus, the hybrid welding speed is decreased reasonably to ensure the accepted weld formation compared with that in single laser welding, but it is still much higher than that for the conventional MIG. From Figure 5, it is indicated that the keyhole depth also fluctuates during the hybrid welding process. Nevertheless, owing to increased heat input and the relatively lower welding velocity, the molten pool volume has a growth in hybrid welding compared to that in MIG. Meanwhile, the basic fluid flow pattern within the molten pool also changes largely due to differences in action forces and the existence of filler metal.

In Figure 5a, it is illustrated that at *t* = 0.0063 s, the molten pool behavior is primarily characterized by laser welding, and a keyhole also appears. As a result of the high thermal conductivity of copper alloy, the influence of arc heat input is not obvious at this moment. At *t* = 0.0102 s, the liquid metal pool volume increases, and the keyhole penetrates the workpiece fully. The droplet also transferred into the weld pool. Similar to that in laser welding, for the front wall of the keyhole, the liquid metal flows downwards at high velocity all the time. However, for the rear keyhole wall, the molten metal also moves downwards, which differs from that in laser welding. The reason for this behavior is that beside the recoil pressure, there exist several other forces acting on the rear keyhole wall, including arc pressure, droplet impingement force, and electromagnetic force, which impel the liquid metal to travel downward to a great extent. In addition, under the condition of a big melt pool, the increased hydrostatic pressure is also conductive to this phenomenon. Thus, the flow behavior of the liquid metal near the rear keyhole wall results from the dynamic competition of several different forces. This behavior benefits the suppression of underfill defect (or shrinkage ditch) occurring at the weld root by providing more liquid metal to some degree. In Figure 5c, it is seen that the melt pool volume increases further due to the effect of arc heat input at *t* = 0.0201 s. Meanwhile, similar to that in laser welding or partial penetration hybrid welding, the keyhole also collapses in full penetration hybrid welding. As mentioned above, this is because the recoil pressure near the keyhole exit at the workpiece bottom has a sharp decrease in penetrated keyhole welding compared with that in non-penetrating welding mode, causing the increased instability of the keyhole lower part.

When the time reaches 0.023 s, the keyhole penetrates the molten pool again. At the same time, with the weld pool volume rising, a clockwise vortex is generated in its middle and rear parts, where the arc dominates the melt pool shape and size. This behavior feature is closely related to the strong downwards flow near the rear keyhole wall and also raises the instability of the rear keyhole wall to some degree, making the keyhole collapse easier. Similar to that in laser welding, in the case of relatively high welding speed, owing to the influences of recoil pressure and Marangoni force, the liquid metal at the underside of the workpiece also flows backwards, causing the occurrence of the melt pool dragging phenomenon. Meanwhile, a small bulge of liquid metal emerges at this domain, which has a certain growth with time due to the continuous accumulation of liquid metal, as seen in Figure 5e. Owing to the strong backward flow, a clockwise vortex is also generated in the weld pool region at the workpiece bottom, which is different from that in non-penetrating hybrid welding. The detailed flow pattern can be observed clearly in the zoomed-in weld pool at *t* = 0.502 s, as shown in Figure 5g. However, due to effect of gravity and relatively high welding speed, the liquid metal bulge at the workpiece back is relatively small. In Figure 5f,g, it is demonstrated that in this condition, the weld reinforcements at the top and bottom sides of workpiece are smooth and no humping and underfilling phenomena happen at the weld root.

Figure 6 shows the temperature field and flow field at the cross-section of the melt pool for different times. It is seen that in the early stage, laser plays a leading role, and the molten pool dynamic behavior is similar to that in laser welding. With the hybrid heat source traveling, the influence of the arc increases gradually, making the molten pool feature close to that in MIG. In Figure 6a, it is observed that at *t* = 0.2065 s, the laser beam moves to this location. The melt pool width is small, within which a non-penetrating keyhole is formed. The liquid metal near the keyhole wall flows toward the molten pool bottom. With the time going by, the keyhole depth is raised quickly and a fully penetrated keyhole appears, as seen in Figure 6c,d. At this stage, the liquid metal still has high velocity of downward flow. Then, since the recoil pressure is reduced significantly at the keyhole exit in the case of the penetrated keyhole, the lower part of keyhole is closed under the action of the surface tension, as shown in Figure 6e. As the heat source travels, the effect of recoil pressure also diminishes further. The keyhole begins to be filled by the liquid metal. However, because of the influence of the downward momentum of liquid metal near the rear keyhole wall and arc pressure as well as gravity, a downward fluid flow can still be observed at this time. However, the flow velocity has a certain decrease. Meanwhile, a metal bulge is also produced at the workpiece back. At *t* = 0.2616 s, the influence of arc heat input is enhanced, leading to an increase in the molten pool width. The molten metal bulge at the bottom surface of the workpiece also grows in size. Under the action of surface tension, the liquid metal flowing downwards at the middle part of the weld pool is redirected toward both sides at the workpiece bottom, as illustrated in Figure 6g. Meanwhile, a small vortex occurs at each side. This flow pattern helps enlarge the contact area of the liquid metal and solid interface, thus also inhibiting the dripping of liquid metal to some degree. Thus, compared with that in laser welding, this relatively stable metal bulge tends to reduce the risk of generating the defect at the weld root. At *t* = 0.3252 s, the arc starts to play a leading role. The molten pool volume has a substantial growth, but the basic flow pattern has no obvious change, as indicated in Figure 6h. Driven by arc pressure and droplet impingement force, a depression is still generated on the upper surface of the weld pool. Two vortexes can also be clearly observed at the lower part. In Figure 6i,j, it is seen that with the arc heat source moving forwards further, the distortion of the weld pool upper surface disappears gradually, and the liquid metal bulge emerges at this domain, which will turn into the weld reinforcement after solicitation. Meanwhile, the fluid flow velocity is also decreased largely with distance away from the arc and laser force sources.

In addition, Figure 6 also indicates that a spattering phenomenon still occurs at the weld pool bottom surface in full penetration hybrid welding, meaning that the arc acting on the workpiece top surface has a minor influence on this defect.

Figure 7 compares the evolution processes of the keyhole depth between the laser and hybrid welding processes. Although the keyhole still has a dynamic fluctuation in full penetration hybrid welding, its collapse frequency is reduced greatly compared with that in single laser welding, thereby contributing to the suppression of welding defects and the resultant improvement of weld formation at the workpiece backside. As expected, this phenomenon should be firstly ascribed to the preheating of the base metal by the arc. In hybrid welding, the laser acts in the liquid metal pool caused by arc directly, and the evaporation of the metal needs less laser energy compared to that in single laser welding. Thus, the keyhole is generated more easily in hybrid welding. Meanwhile, as mentioned above, both arc pressure and droplet impingement force strengthen the downward flow of the liquid metal on the rear keyhole wall, which also makes some contributions to the keyhole stability at the weld root. In addition, it can also be found in Figure 7 that the maximum variation magnitude of keyhole depth in hybrid welding is still similar to that in single laser welding, demonstrating that the arc and filler metal cannot affect the collapse position of keyhole effectively. This is due to the formation and maintenance of the keyhole being mainly ascribed to the recoil pressure induced by the laser. In the case of a bent keyhole, the laser energy has no relatively stable distribution at the keyhole lower part, where the keyhole collapse usually occurs. Although the arc and filler metal can improve the flow pattern of liquid metal near the rear keyhole wall, they are still not the critical factors responsible for maintaining the keyhole.

Therefore, in addition to the positive effect, the addition of the arc and fill metal also has a certain negative influence on the weld formation in hybrid welding of copper alloy. As reported by Wu et al. [26] and Xu et al. [30], the forward metal flow induced by the clockwise vortex makes the hydrodynamic stress exerting on the rear keyhole wall more complex, being adverse to the keyhole stability. However, different from that in partial penetration hybrid welding, the liquid metal can flow backwards at the workpiece backside in the case of full penetration, which is able to offset the negative effect partially. On the one hand, although a large molten pool volume is conductive to the suppression of the weld surface depression and underfill defects, it raises the potential threat of generating the other welding defects to some degree, including weld pool collapse and burn through, which usually happen in the arc welding of a thin plate and are difficult to be controlled. However, a high moving speed of the heat source in hybrid welding offers the opportunity to solve this issue well compared with that in conventional arc welding. Moreover, the high thermal conductivity of copper alloy also hinders the occurrence of these defects to some degree. However, obtaining the acceptable weld formation still needs the reasonable matching of different welding parameters. In the future, the further simulation study involving the influence of other different welding parameters and the process optimization will be performed.

In order to verify the accuracy of the above model, the weld cross-section geometry and dimensions calculated under different welding conditions were also compared with the experimental results, as shown in Figure 8. Both are in general agreement, but there is still some discrepancy in the shape of the weld. The reason for the error between them is related to the simplified model and the lack of accurate thermal properties of materials at different temperatures, which will also be solved in our future research.

## 5. Conclusions

(1)A three-dimensional numerical model is developed to investigate the weld formation in full penetration laser + MIG hybrid butt welding of thin copper alloy, which can calculate the temperature and flow fields, keyhole behavior, as well as the weld defects. The validity of the model is verified by comparing the computed results with the experimental data.(2)In single laser welding, the molten pool volume is small due to the energy concentration and high welding speed. No vortex can be formed within the tiny weld pool. Due to the lack of direct irradiation of a laser beam and the resultant steady recoil pressure, the flow pattern near the rear wall is relatively complex, leading to the drastic fluctuation of keyhole. In the condition of this study, a severe spatter phenomenon emerges at the backside and the underfill defect is generated in both the upper and lower sides of final weld bead, which are mainly attributed to the recoil pressure and strong downward flow.(3)In hybrid welding, the metal pool has a relatively large volume due to the addition of a heat input, and a liquid metal bulge is formed at the backside. The downward flow of liquid metal near the rear keyhole wall is strengthened under the combined effect of arc pressure, electromagnetic force, and droplet force. For the weld pool longitudinal section, there are clockwise vortexes appearing at both the upper and lower parts. At the cross-section, a vortex is also produced at each side of the central axis near the lower part.(4)Compared with that in laser welding, the keyhole stability is improved in hybrid welding and the tendency of metal bulge dripping at the weld pool back is also reduced to some extent under the effect of the vortex occurring at the weld pool lower part. Thus, the formation defect can be suppressed effectively at the top and bottom sides of the weld bead. The spattering phenomenon still occurs at the workpiece bottom, showing that the addition of arc and filler metal has a minor influence on this defect.

## Figures and Tables

**Figure 1 materials-13-05307-f001:**
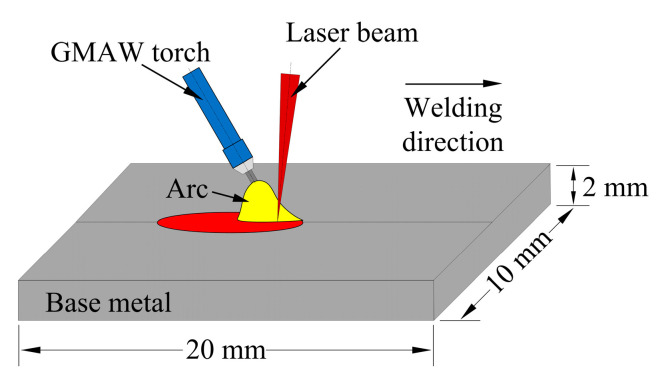
Schematic of laser + MIG hybrid welding.

**Figure 2 materials-13-05307-f002:**
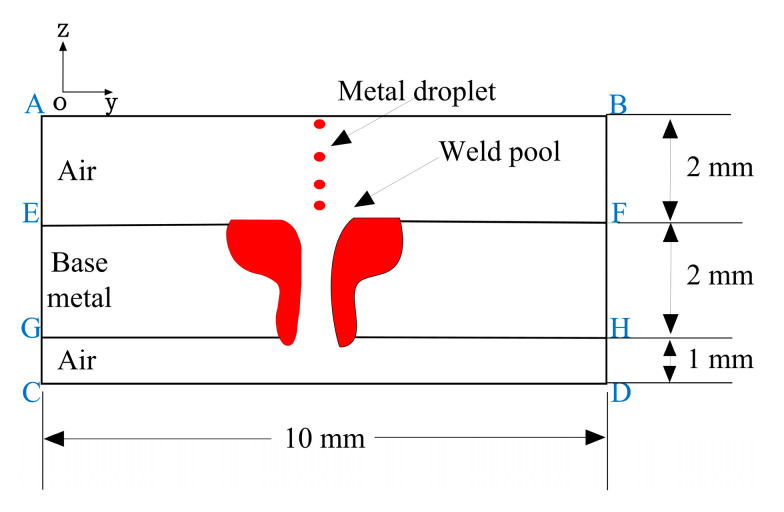
Schematic of calculation domain.

**Figure 3 materials-13-05307-f003:**
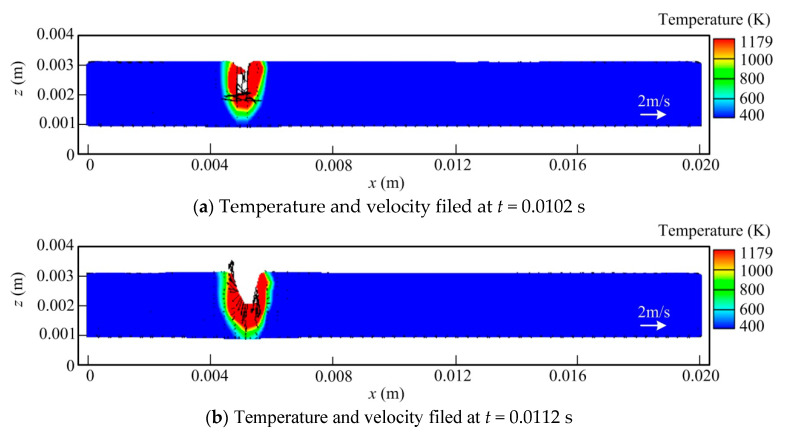
Evolution of temperature and velocity fields at longitudinal section in laser welding (m: zoomed-in weld pool at *t* = 0.085 s).

**Figure 4 materials-13-05307-f004:**
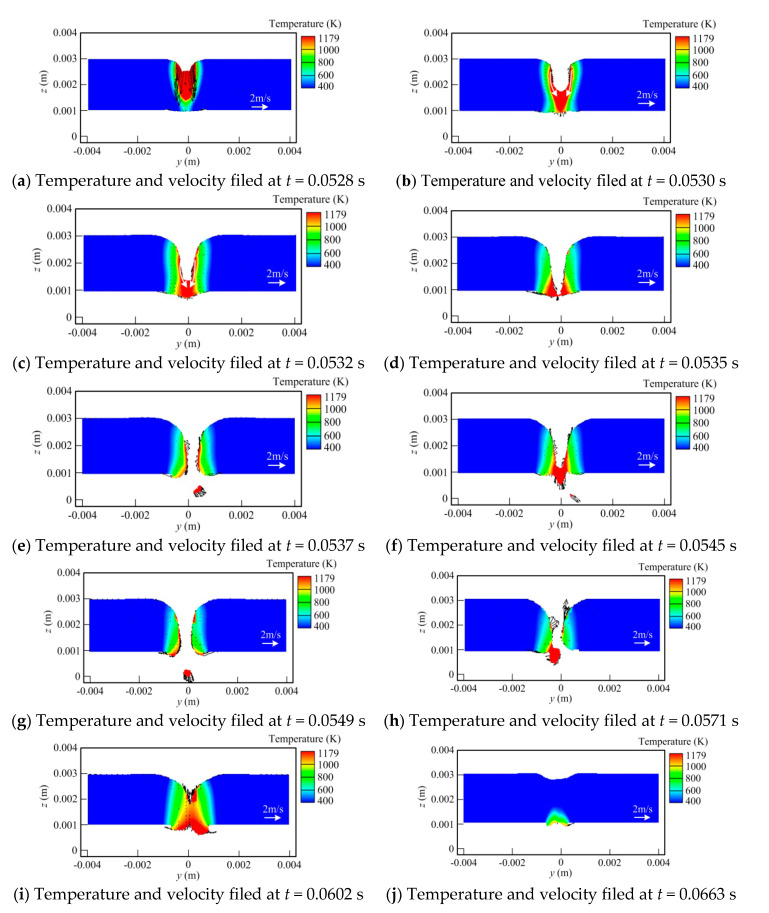
Evolution of temperature and velocity field at cross-section in laser welding (x = 0.010 m).

**Figure 5 materials-13-05307-f005:**
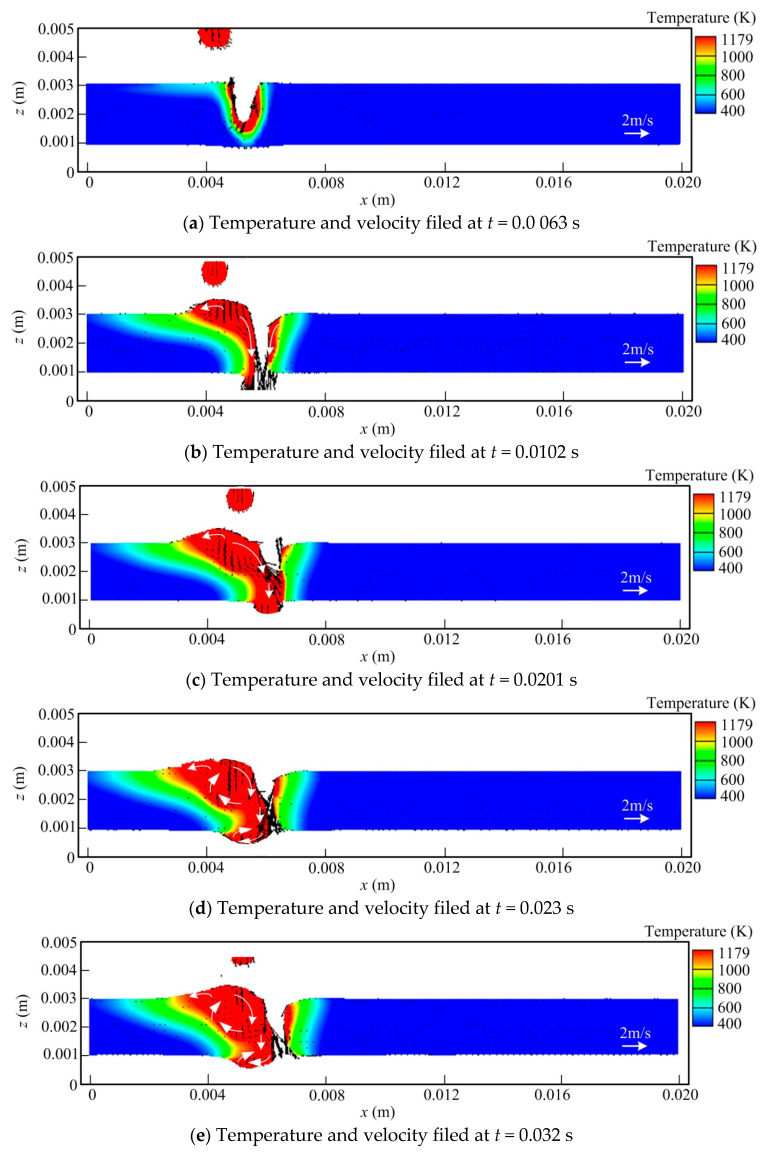
Evolution of temperature and velocity field at longitudinal section in laser welding (g: zoomed-in weld pool at *t* = 0.502 s).

**Figure 6 materials-13-05307-f006:**
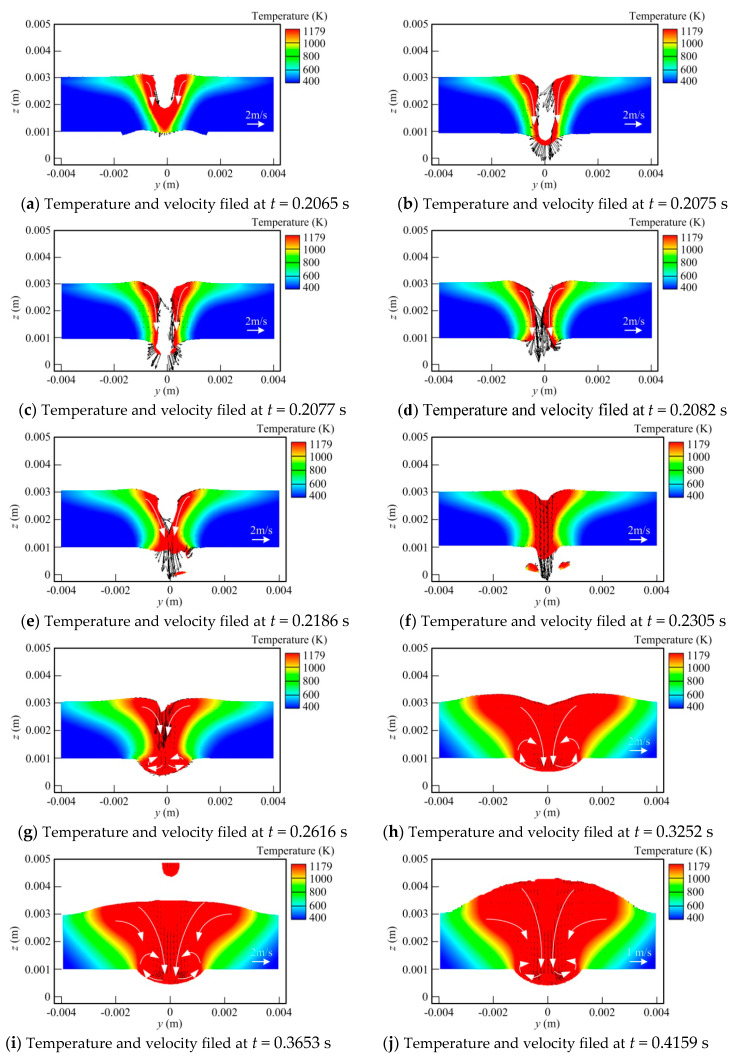
Evolution of temperature and velocity field at cross-section in hybrid welding (x = 0.008 m).

**Figure 7 materials-13-05307-f007:**
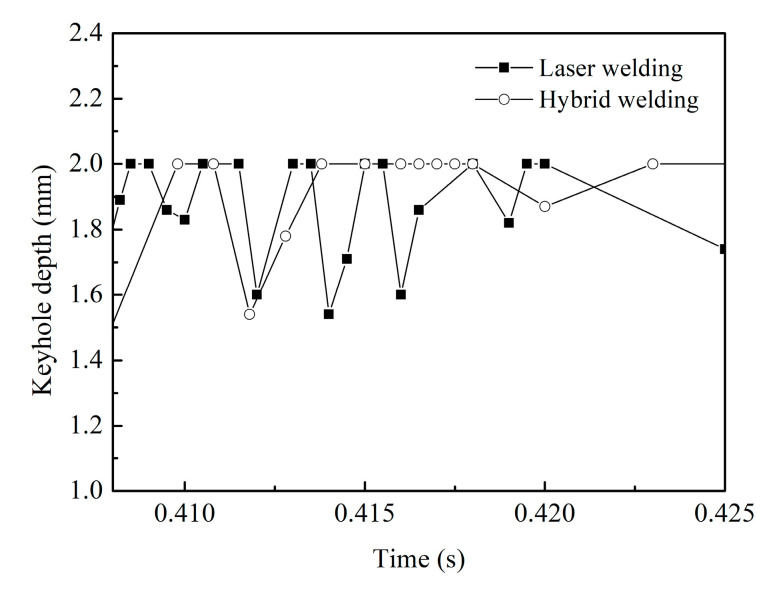
Evolution of keyhole depth for different welding conditions.

**Figure 8 materials-13-05307-f008:**
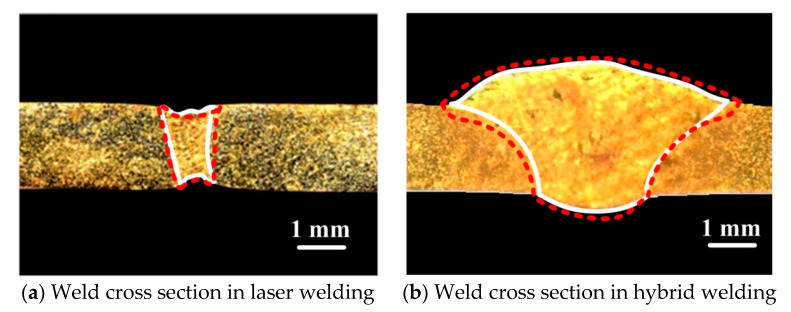
Comparison of calculated cross-section with the experimental data (white and red lines represents the experimental and calculated results, respectively).

**Table 1 materials-13-05307-t001:** Chemical compositions of S221 copper alloy wire (wt%).

Cu	Sn	Si	Zn
60	1.0	0.3	Bal.

**Table 2 materials-13-05307-t002:** Process parameters in laser + MIG (metal inert gas) hybrid welding.

Case No.	Welding Current (A)	Welding Speed (m s^−1^)	Laser Power (kW)
1	0	0.035	3
2	180	0.030	3

**Table 3 materials-13-05307-t003:** Thermo-physical properties and other parameters used in calculation_0._

Material Properties	Value
Latent heat of fusion, *L**_m_*	1.88 × 10^5^ J/kg
Latent heat of evaporation, *L_b_*	5.44 × 10^6^ J/kg
Thermal expansion, *β*	2.06 × 10^−5^/K
Solidus temperature, *Ts*	1179 K
Liquidus temperature, *T*_l_	1179 K
Ambient temperature, *T*_0_	293 K
Stefan–Boltzmann constant, *σ*	5.57 × 10^−8^ W/(m^2^K^4^)
Surface tension coefficient, *γ*	1.3 N/m
Dynamic viscosity of liquid phase, *μ*	0.0041 kg/(m^3^ s)
Thermal conductivity of solid phase, *k**_s_*	381 W m^−1^K^−1^
Thermal conductivity of liquid phase, *k**_l_*	308 W m^−1^K^−1^
Specific heat of liquid phase, *c**_pl_*	450 J kg^−1^K^−1^
Specific heat of solid phase, *c**_ps_*	384 J kg^−1^K^−1^
Density of copper alloy, *ρ*	8430 kg/m^3^

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
