# Peer review of "Simulation Study on Weld Formation in Full Penetration Laser + MIG Hybrid Welding of Copper Alloy"

_materials, 2020, doi:10.3390/ma13235307_

Round 1

Reviewer 1 Report

Thank you for submitting this manuscript. This is great work that will be of particular interest to welding engineers or other experts really applying the technology.

I think the manuscript is good and I only want to recommend a few changes that might further improve the quality of your work:

  • make sure there is a space between the references and text

Abstract:

  • use a less complicated/long first sentence for a quick start into this

Introduction:

  • You can use a figure in the introduction to describe the current challenges easier
  • You can put the different studies you list (good work) for the state of the art in a table so that this is more concise

Experimental:

  • You can provide a picture of your actual set-up next to the schematic in Figure 1 - you did experimental work and it will be for engineers. Pictures can help a lot here.

Author Response

[1] Make sure there is a space between the references and text.

Response: We are thankful to the reviewer for the careful observation and suggestions. According to the review comment, a space has been added between the reference and the text.

[2] Use a less complicated/long first sentence for a quick start into this Introduction.

Response: We are very thankful to the reviewer for the good suggestions. According to the review comment, the first sentence has been revised and becomes concise, which is highlight.

[3] You can use a figure in the introduction to describe the current challenges easier. You can put the different studies you list (good work) for the state of the art in a table so that this is more concise

Response: We are very thankful to the reviewer for the good suggestions. Fig.8(a) can show the formation defect usually occurring in full penetration laser welding of copper alloy. It is a good suggestion to list the previous research works using a table, which can make the introduction to the state of art more concise. But, we are afraid that we cannot describe the relationship and difference among previous studies clearly in this reseach. In the future study, we will try to depict the state of the art with a table.

[4] You can provide a picture of your actual set-up next to the schematic in Figure 1 - you did experimental work and it will be for engineers. Pictures can help a lot here.

Response: We are very thankful to the reviewer for the good suggestions. To make the set-up schematic clearer, Fig.1 has been revised, which is highlighted. Due to that the pictures of our actual set-up may not reach the quality requirement of the journal, we are sorry for that we cannot provide the picture with high quality in this manuscript. In the future study, we will prepare the picture of actual set-up with high quality timely.  

Reviewer 2 Report

The reviewed manuscript requires a major correction of a native speaker (English). There are many mistakes not only grammatical but also linguistic mistakes. The title of the paper should be improved. Moreover there are many technical mistakes, e.g. word spacing is many times incorrect. All this causes that the review article is hard to read. Authors should insert an introductory sentence in chapter 3 "Modelling". The reviewed article can be published in Materials journal only after mayor corrections. All necessary corrections are contained in the attached file.

Author Response

[1]The reviewed manuscript requires a major correction of a native speaker (English). There are many mistakes not only grammatical but also linguistic mistakes.

Response: We are thankful to the reviewer for the careful observation and suggestions. The writing and technical mistakes have been checked and the errors have been corrected, which are lighted in the revised manuscript.

[2]The title of the paper should be improved

Response: We agree with the review comment. The tile has been revised and is concise.

[3] Authors should insert an introductory sentence in chapter 3 "Modelling".

Response: According to the review comment, an introductory sentence has been added in the modeling section (P4, lines1-2) in the revise manuscript, which is highlighted.

Reviewer 3 Report

I feel embarrassed with this paper for different reasons : 

The paper deals with the model developed by Guogang Xiu - cited as  ref 14 - below.  

In the paper you are applying this model to Cu alloy and I think go a little bit deeper / longer ? in the calculation/interpretation of the calculus : this is not "clear" 

The results are different from Ref 14.  and may be interesting but suffer of  either   "weaknesses" or awkwardness : 

  • while not "truly" plagiarism,  line 124 to 268 - "presentation of the model" are nearly/totally the same as those in ref 14.   And nearly the same for Fig 1. and Fig 2.   Sometime it could be interesting for the ease/convenience of reading to repeat some details from another paper  but there, this is clearly too much ! ! !   In this case - just say after the introduction that in this paper you are employing the model developed in ref 14, to Cu.  

  • in Table 3 - the Latent Heat of fusion. / evaporation are false - two times : the exponent is 105 of course and not 10-5 (this could be just a minor typing error) but also the values are the same as the one in ref. 14 - i.e. those for Al, and not for Cu for which the paper is supposed to deal with : is it a bad copy/ paste or is it a real error (other properties looks more coherent with Cu alloys...) 
  • If we believe that it's a typing error -( How can we be sure of that ? ), then, going through the results, one can not really know or understand :
    • how is chosen the section presented   i.e. why the section in Fig 3. are chosen at non constant time interval ?  Is it due to the fact that those are more "representativ " representative with regards to what ?   Why should I believe you ? 
    • why the time chosen for Fig 4. are not the same as for Fig 3. with the same remarks concerning the choice of the section,
    • Where is located - along the x  axis the "y-z" section ?  How do you chose this location ? Why ? 
    • are "all the figure" useful ? 
    • The black arrow are non readable in the figure : please make a zoom at least for one or two.  
  • the same comments for  Fig. 5 and 6.  
  • In general - what about the uncertainty ? 
  • Fig. 7 must be clarified and more explanation given.  
  • The conclusions looks in relation with the paper but must be deepened instead of "further work"... 

Regards.  

Author Response

 [1] The paper deals with the model developed by Guogang Xiu - cited as ref 14 - below. In the paper you are applying this model to Cu alloy and I think go a little bit deeper / longer in the calculation/interpretation of the calculus : this is not "clear".

Response: I agree with the reviewer comment. Different from that in Ref.14, the model can be used to calculate the temperature field, keyhole behavior and fluid flow in full penetration hybrid welding process, which are more complicated than those in the partial penetration welding reported in Ref.14. This is due to that, at the back side of workpiece, part of laser beam can leave the base metal through the keyhole exit. The thermal and pressure conditions have larger transient changes in the case of full penetration and penetrated keyhole. Thus, in this condition, the heat source model and recoil pressure distribution have to been modified to some extent when calculating the thermal and fluid flow at the backside. Meanwhile, the coupling of gas, liquid and solid phases at the backside also increases the difficulty in convergence of solution. According to the comment, the related explanations have been added to the revised manuscript at the section (lines ).

[2] The results are different from Ref 14.  and may be interesting but suffer of  either   "weaknesses" or awkwardness : while not "truly" plagiarism,  

Response: Due to differences in material properties and welding process (full penetration and partial penetration), the characteristics of thermal field, keyhole behavior and fluid flow as well as the weld defect formation process in this study are different from those in Ref.14. In this study, we mainly focused on the investigation of keyhole behavior and fluid flow though the simulation method in full penetration welding, epically for that in the case of penetrated keyhole. Besides, the underfill defect and spatter are also studied in this study, which are not involved in our previous study (Ref.14)

To investigate the evolution of keyhole and weld pool, the simulated results at different times are shown in the manuscript. It should be noted that, before the keyhole penetrates the workpiece fully, it is still a blind keyhole. At this moment, keyhole and weld pool behavior are similar to those during the partial penetration welding process, as reported in Ref.14. Therefore, the explanations are also similar. But, when a full penetrated keyhole is generated, the keyhole geometry and behavior as well as fluid flow pattern will change largely. Spatter defects also emerges, which do not occurs in the work of Ref.14. Meanwhile, the physical phenomenon at backside of weld pool is also the focus of this research, which is not concerned in Ref.14. 

    Meanwhile, we agree with the review comment. This manuscript is also revised, making the difference clear, which is highlighted.

[3] line 124 to 268 - "presentation of the model" are nearly/totally the same as those in ref 14.   And nearly the same for Fig 1. and Fig 2.   Sometime it could be interesting for the ease/convenience of reading to repeat some details from another paper  but there, this is clearly too much ! ! !   In this case - just say after the introduction that in this paper you are employing the model developed in ref 14 to Cu.

Response: We are thankful to the careful observation and suggestions. We agree with the comment. According to the comments, the related presentations have been revised at the model section. In the simulation study on weld pool behavior, all the basic governing equations and the basic thermal and pressure conditions as well as the basic calculation prese are the same or close to each other, which can be found in lots of literatures, such as F,, . The difference mainly shows in the distribution mode of heat source, arc or laser induced pressure, grid system, resolution mechanism, coupling mechanism, et ac. As mentioned above, in this manuscript, the model is extended to the full penetration case and through dealing with the boundary conditions at the backside of work piece. Also, both Fig.1 and Fig.2 are revised.

[4] in Table 3 - the Latent Heat of fusion. / evaporation are false - two times : the exponent is 105 of course and not 10-5 (this could be just a minor typing error) but also the values are the same as the one in ref. 14 - i.e. those for Al, and not for Cu for which the paper is supposed to deal with : is it a bad copy/ paste or is it a real error (other properties looks more coherent with Cu alloys...) .If we believe that it's a typing error -( How can we be sure of that ? ), then, going through the results, one can not really know or understand :

Response: We are thankful to the carful observation and agree with the review comment. It is all our fault and a bad copy is used. Some material properties in Table 3 are false, which really results from the typing error. All the material properties have been checked and revised. Ref.14 is our previous work. Due to that this table in the manuscript is made base on the table in Ref.14, some properties are forgotten to change when writing the manuscript. But, in the calculation, the correct properties are used. We thank the reviewer again for the careful .The comparison of simulated and experimental results can also validate the accuracy of the material properties.

[5] how is chosen the section presented   i.e. why the section in Fig 3. are chosen at non constant time interval ?  Is it due to the fact that those are more "representativ " representative with regards to what ?   Why should I believe you ? why the time chosen for Fig 4. are not the same as for Fig 3. with the same remarks concerning the choice of the section,

Response: Due to that the laser energy and recoil pressure distribution are related to the keyhole behavior, which are always in dynamic change. Meanwhile, droplet also impinges weld pool in welding. Thus, the weld pool and keyhole dynamic behavior are relatively complicated and cannot reach the steady state. To show the keyhole clearly, the central longitudinal section of weldment (i.e. the symmetric plane) is usually selected. The hybrid heat source center is located at this section and the longitudinal view of keyhole and weld pool can be depicted fully. In the simulation study, this section is widely chosen (such as Wu C S, et al, Journal of Materials Manufacturing, 25:235-245 ) .

Meanwhile, due complexity of keyhole and weld pool behavior, some typical phenomenon cannot be observed when choosing the constant time interval, i.e., the time interval of opening and closure for keyhole at the backside or occurrence of spatter are not fixed. Besides, the calculated results cannot be saved for the entire time step by the computer due to limitation of memory. Therefore, to reveal the fluctuation process of keyhole, weld pool and spatters fully, a variable time interval has to been used.

Besides, it takes little time for weld pool and keyhole to go through a cross section due to high welding speed and relatively small size of weld pool and keyhole. For a certain cross section, some typical phenomena of weld pool and keyhole may not be observed while choosing the same interval, such as the formation process of keyhole opening and spatter at the backside of weld pool. Thus, it has to choose different time interval for the longitudinal and cross-sectional weld pool and keyhole calculated results sometimes(such as Cho, et al, Welding Journal, 2009, 88:35-43).

[6] Where is located - along the x axis the "y-z" section ? How do you chose this location ? Why ?

are "all the figure" useful ?

Response: The location of the cross section has been given in the caption of the figture. As mentioned above, during welding process, the weld pool and keyhole cannot reach the real steady state and are always in change. To reveal the welding physical process, we choose some locations, where the weld pool and keyhole behavior features are more obvious, helping to show the transient evolution process clearly. Besides, due to that the calculated results cannot be saved by the computer for each time step, we have to choose the cross section which can demonstrate the weld pool behavior feature clearly. Thus, this method is widely used in the simulation study.  

[7] The black arrow are non readable in the figure : please make a zoom at least for one or two.  

the same comments for  Fig. 5 and 6.  

Response: We are thankful to the reviewer for the good suggestions. During hybrid welding, the flow velocity of liquid metal near the keyhole is much higher than that in the other region. Therefore, the length of black arrow is much longer. When a suitable length of black arrow for liquid metal near the keyhole wall is used, the length of arrow representing the velocity of liquid metal inside the weld pool may be small and non readable. The related revision has been made (Fig.3(k) and Fig.5(m)), which are highlighted.

 [8] Fig. 7 must be clarified and more explanation given.  

Response: According to the review comment, Fig.7 has been clarified and the related explanation has also been added in the revised manuscript, which is highlighted.

[9] The conclusions looks in relation with the paper but must be deepened instead of "further work"... 

Repsonse: We agree with the review comment. The conclusion has been revised, which is more deepened. The contents concerning the further work have been deleted. The related revision has been highlighted.